# 1,3,4-Oxadiazole Derivative Attenuates Chronic Constriction Injury Induced Neuropathic Pain: A Computational, Behavioral, and Molecular Approach

**DOI:** 10.3390/brainsci10100731

**Published:** 2020-10-13

**Authors:** Muhammad Faheem, Syed Hussain Ali, Abdul Waheed Khan, Mahboob Alam, Umair Ilyas, Muhammad Zahoor, Muhammad Umar Khayam Sahibzada, Sidra Khalid, Riaz Ullah, Ali S. Alqahtani, Abdulaziz M. Alqahtani

**Affiliations:** 1Riphah Institute of Pharmaceutical Sciences, Riphah International University, Islamabad 44000, Pakistan; hussainali.sa@gmail.com (S.H.A.); umair.ilyas@riphah.edu.pk (U.I.); 2Department of Pharmacy, The University of Lahore, Islamabad 44000, Pakistan; waheedmarwat31@gmail.com; 3Department of Pharmacy, Capital University of Science and technology, Islamabad 44000, Pakistan; mahboob.alam@cust.edu.pk; 4Department of Biochemistry, University of Malakand, Chakdara, Dir (Lower), KP 23050, Pakistan; 5Department of Pharmacy, Sarhad University of Science and Information Technology, Peshawar, KP 25000, Pakistan; umar.sahibzada@gmail.com; 6Department of Pharmacy, Faculty of Biological Sciences, Quaid-i-AzamUniversity, Islamabad 44000, Pakistan; sidra.merlin@gmail.com; 7Department of Pharmacognosy, College of Pharmacy, King Saud University, Riyadh 11451, Saudi Arabia; rullah@ksu.edu.sa (R.U.); alalqahtani@ksu.edu.sa (A.S.A.); 439106051@ksu.edu.sa (A.M.A.)

**Keywords:** In-silico, chronic constriction injury, hyperalgesia, neuro-protective, IL-6

## Abstract

The production and up-regulation of inflammatory mediators are contributing factors for the development and maintenance of neuropathic pain. In the present study, the post-treatment of synthetic 1,3,4 oxadiazole derivative (B3) for its neuroprotective potential in chronic constriction injury-induced neuropathic pain was applied. In-silico studies were carried out through Auto Dock, PyRx, and DSV to obtain the possible binding and interactions of the ligands (B3) with COX-2, IL-6, and iNOS. The sciatic nerve of the anesthetized rat was constricted with sutures 3/0. Treatment with 1,3,4-oxadiazole derivative was started a day after surgery and continued until the 14th day. All behavioral studies were executed on day 0, 3rd, 7th, 10th, and 14th. The sciatic nerve and spinal cord were collected for further molecular analysis. The interactions in the form of hydrogen bonding stabilizes the ligand target complex. B3 showed three hydrogen bonds with IL-6. B3, in addition to correcting paw posture/deformation induced by CCI, attenuates hyperalgesia (*p <* 0.001) and allodynia (*p <* 0.001). B3 significantly raised the level of GST and GSH in both the sciatic nerve and spinal cord and reduced the LPO and iNOS (*p <* 0.001). B3 attenuates the pathological changes induced by nerve injury, which was confirmed by H&E staining and IHC examination. B3 down-regulates the over-expression of the inflammatory mediator IL-6 and hence provides neuroprotective effects in CCI-induced pain. The results demonstrate that B3 possess anti-nociceptive and anti-hyperalgesic effects and thus minimizes pain perception and inflammation. The possible underlying mechanism for the neuroprotective effect of B3 probably may be mediated through IL-6.

## 1. Introduction

Peripheral nerve injury is more prevalent in traffic and industrial accidents, acts of violence, etc. The defects observed so far are usually mixed motor and sensory in nature. This leads to an increase in global nerve repair and regeneration costs from $4.5 billion in 2013 to $7.8 billion in 2018. Thus, major innovations in the design of treatment and to lower the overall cost of the therapy are needed [1]. Different types of inflammatory mediators (IFMs) are involved in the development of neuropathic pain including interleukin-6 (IL-6), cyclooxygenase-2 (COX-2), and nitric oxide synthase (NOS) [2]. IL-6 is an extensively studied cytokine which is involved in neuro-inflammation [3]. It exerts its biological effects through membrane-bound interleukin-6 receptors (mIL-6R) [4]. The complex of IL-6 and mIL-6R then activates the downstream proteins termed gp130, which have been found to be expressed in almost all kinds of cells while mIL-6R is specifically present on the surface of neutrophils, leukocytes, macrophages, and monocytes [5]. The gp130, upon activation, results in the stimulation of mitogen-activated protein kinase (MAPK). MAPK is then involved in the development of pain and increases the sensitivity to mechanical and thermal stimuli, causing hyperalgesia and allodynia [6]. The current research work is concerned with the pharmacological investigation of 1,3,4-oxadiazole (Figure 1) derivative (B3) in chronic constriction injury (CCI) induced neuropathy. B3 has been already investigated for its analgesic and anti-inflammatory [7] potential utilizing different animal models.

This research utilizes the computational, behavioral, and molecular approaches to determine the curative potential of the B3 in CCI-induced neuropathic pain. 

## 2. Materials and Methods

### 2.1. Chemicals

Reduced glutathione (GSH), GST, Trichloroacetic acid (TCA), Cholrodinitro benzene (CDNB), Dihydrodithiobisnitro benzoic acid (DTNB), ketamine, lipid peroxidase (LPO), xylazine, dimethyl sulphoxide (DMSO), and IL-6 ELIZA kit, and 1,3,4-oxadiazole derivative were utilized in this study. The chemical used were of analytical grade.

### 2.2. Animals

Sprague Dawley rats weighing 180–250 g were utilized in the study. The animals were given free access to a standard diet and water ad libitum with a 12-h light–dark cycle and kept in a controlled environment of temperature (25–30 °C) and humidity. The research work was performed in accordance with the research and ethical committee of Riphah Institute of Pharmaceutical Sciences, Riphah International University, Pakistan, with an approval number Ref. No. REC/RIPS/2020/20.

### 2.3. Animal Grouping and Dosing

The animals were divided into five groups with each group having 6 rats.

Saline: Group I served as the control (saline, 10 mL/kg body weight) without surgical procedure.

Sham: Group II served as a sham (saline 10 mL/kg body weight) with surgical procedure and without the constriction of the sciatic nerve.

CCI + B3: Group III and IV were the treated groups, administered with 10 and 20 mg/kg body weight of B3 with surgical procedure and constriction of the sciatic nerve.

CCI: Group V served as a disease group (CCI) with surgical procedure and constriction of the sciatic nerve. The scheme of the study is given in Figure 2.

The surgery was performed initially for certain groups. The dosing started one day after the surgery, until the 14th day. Behavioral studies were conducted before surgery on day 0 and after surgery on day 0, 3rd, 7th, 10th, and 14th. After 14th day the animals were sacrificed, and tissue samples (sciatic nerve and spinal cord) were collected.

### 2.4. Docking Studies

The docking studies of 1,3,4-oxadiazole derivative were carried out through Auto Dock Vina and PyRX software [8] against targets involved in neuropathic pain including iNOS (PDB ID: 4D31), COX-2 (PDB ID: ICX2) and IL-6 (PDB ID: 2Q34) as shown in Figure 3. All the targets were downloaded from protein data bank (http://www.rcsb.org/pdb/home/home.do) in PDB format purified through the “Discovery Studio Visualizer” (DSV). Binding energy value expressed in kcal/mol of the best pose of the ligand–receptor complex was obtained through PyRx software. DSV was also used to prepare a 3D structure of the ligand B3 and saved in PDB format as shown in (Figure 1b). The post-docking analysis was carried out to schematically represent hydrogen bonding (classical and non-classical), hydrophobic interactions, and amino acid residues involved in hydrogen bonding of the best-docked pose of the ligand–protein complex [8].

### 2.5. Sciatic Nerve Constriction

The sciatic nerve constriction was performed by the methodology described in a previous study [9]. In accordance with this method, the rats were anesthetized with ketamine (10 mg/kg body weight) and xylazine (100 mg/kg body weight) and the sciatic nerve was constricted with four loose ligatures 1 mm apart with sutures (3/0). Treatment with B3 was started one day after surgery and continued until the 14th day. All behavioral studies were executed on day 0, 3rd, 7th, 10th, and 14th. The animals were sacrificed and tissue samples (sciatic nerve and spinal cord) were collected for further analysis.

### 2.6. Behavioral Studies

#### 2.6.1. Thermal Hyperalgesia

After the habituation of the animals, thermal hyperalgesia was assessed by placing the hind paw on a hot plate set at a temperature (54 ± 1 °C). Cut off time was 20 s. Six consecutive reading with an interval of 10 min were taken [10].

#### 2.6.2. Mechanical Allodynia

Von Frey filament/Vonfrey hairs were used to determine the mechanical allodynia [11]. The animals, after acclimatization, were placing on a mesh, the Von Frey filaments (1, 2, 4, 6, 8, 10, and 15 g) was applied on the planter surface from low pressure to high pressure. The force was applied in such a way that the filament was bent for 2–3 s and sudden withdrawal or licking of the paw was noticed.

#### 2.6.3. Paw Deformation

Paw deformation was determined by the methodology explained by Nakazato et al., with slight modifications [12]. In accordance with this method, the hind paw of the rat was dipped in ink and then the rat was allowed to walk on a white piece of paper. The footprint of rats in all groups were obtained and then evaluated for paw deformations.

### 2.7. Biochemical Investigations

#### 2.7.1. LPO Assay

Lipid peroxidation was performed in order to determine the level of Thiobarbituric acid reactive substances (TBARS) by colorimetric method [13]. According to this assay, a 200 µL supernatant of the homogenized samples (brain and spinal cord) was mixed with 580 mL phosphate buffer, 20 µL ferric chloride, and 200 µL ascorbic acid. The mixture was allowed to incubate for an hour at 37 °C. This step was followed by the addition of 1000 µL of each 10% trichloroacetic acid and 0.66% thiobarbituric acid to stop the reaction. The samples were placed in water at room temperature, then immersed in cold water, centrifuged, and the concentration of TBARS was determined at 535 nm expressed in Nm/min/mg protein.

#### 2.7.2. Nitric Oxide Assay

The concentration of NO was determined as per the methodology explained elsewhere [14]. According to this protocol, 50 µL of the supernatant was added to 50 µL and Griess reagent (50 µL). This reagent consists of sulfanilamide (1%), in 0.1% naphthyleethylenedaiaminedihydrochloride, and 5% phosphoric acid, and was then incubated at 37 °C for half an hour and absorbance was measured on a microplate reader (Bio ELx 808). Sodium nitrate solution was used to draw the calibration curve and determine the absorbance coefficient.

#### 2.7.3. Estimation of Oxidative Stress

The determination of oxidative stress markers is important in order to assess the damage caused by the constriction of the sciatic nerve. The tissue samples were homogenized, centrifuged at 4 °C and then the supernatant was collected [15]. GST was determined by mixing 0.2 mL of supernatant with a 2 mL solution of DTNB (0.6 Mm) and sodium phosphate (0.2 M). The required volume was prepared with the addition of 3 mL phosphate buffer, incubated for 10 min, and absorbance was measured at 412 nm using a spectrophotometer. The solution of DTNB and phosphate buffer served as control. GSH level was expressed in µmol/mg of proteins. The level of GST was measured according to the previously described method [16]. Then, 60 µL of the supernatant was added to a freshly prepared solution (5 mM GSH, 1 mM CDNB in 0.1 M phosphate buffer) of 1.2 mL in triplets using glass vials. About 210 µL from the mixture was transferred to a microplate and the reaction rate was determined at 340nm at room temperature using ELIZA microplate reader (Bio TekELx 808, Winooski, VT, USA). The addition of 60 µL water instead of tissue served as blank/control. The GST was measured as µmol of CDNB conjugate/min/mg of proteins.

### 2.8. Hematoxylin and Eosin Staining

The slides containing tissue samples were de-paraffinized and passed through xylene (100%), ethanol (95 and 70%), and distilled water. After washing with PBS, the slides were kept in hematoxylin and in running tap water for 10 and 5 min, respectively. The slides were then exposed to 1% of HCl and ammonia each and immediately rinse with distilled water. They were then immersed in eosin for 5–10 min and rinsed again and dried in the open air. Further dehydration was carried out by a series of serial ethanol dilutions (70%, 95%, and 100%) fixed in xylene and cover with slip. An Olympus light microscope was used for capturing images and analysis of the picture was carried out through ImageJ software [17].

### 2.9. Immuno-Histopathological Evaluation

Immuno-histochemical analysis was performed according to the methodology explained elsewhere [18]. The deparaffinization was performed by washing slides with xylene and serial dilutions (100%, 90%, 80%, and 70%) of ethanol. The slides were then washed with distilled water and then in PBS. The antigen retrieval was done by the addition of proteinase K enzyme and then the slides were washed again in PBS for 10 min. To block the endogenous peroxidase, the slides were treated with hydrogen peroxide at room temperature for 10 min. After this, the slides were washed with PBS and normal goat serum (5%) was applied to each tissue on the slide and incubated for 1.5 h at room temperature. The primary antibody (anti-IL-6) was then applied, and the slides were kept overnight in an incubator at 4 °C. The slides were washed with PBS on the next day and treated with biotinylated secondary antibody (1:50 dilution) and again incubated at room temperature for 2 h. Following washing with PBS, the slides were then subjected to ABC and again incubated in a humidifier for 1 h. After washing with PBS, the slides were stained in 0.1% DAB (diaminobenzidine peroxidase) solution. Upon the completion of staining with the DAB solution, the slides were dehydrated in xylene and serial ethanol dilution and dried in open air. The coverslips were then applied and with the help of mounting media and Tiff images were obtained with a microscope.

### 2.10. ELISA

The expression of inflammatory markers, including IL-6, was determined by using commercially available ELISA kits according to the manufacturer’s instructions. The supernatant of the sciatic nerve and spinal cord was treated with designated antibodies in 96 well plates and the expression level of inflammatory markers (IL-6) was determined by using ELISA microplate reader. Values were expressed as pictograms per milliliter (pg/mL). The procedure was repeated at least three times [19].

### 2.11. Statistical Analysis

The data has been presented as mean ± SEM. H&E staining and oxidative stress data were analyzed using one-way ANOVA and behavioral data was determined by two-way ANOVA, followed by post hoc Bonferroni’s multiple comparison test using GraphPad Prism-6 software. ImageJ software was used to analyze the morphological data. A two-way ANOVA followed by post hoc Bonferroni’s Multiple Comparison test was performed for ELIZA. Symbols # or * represent significant difference values *p* < 0.05, ## or ** represent *p* < 0.01, and ### or *** represents *p* < 0.001 values for significant differences.

## 3. Results

### 3.1. Docking

The ligand molecule (B3) was docked against different targets proteins including IL-6, iNOS, and COX-2. B3 formed three hydrogen bonds against IL-6 with the highest binding affinity of −9.2 kcal/mol **(**Figure 4), one hydrogen bond against NOS with a binding affinity of −6.6 kcal/mol (Figure 5), and two hydrogen bonds against COX-2 with a binding affinity of −7.2 kcal/mol (Figure 6). The number of hydrogen bonds, binding energies (kcal/mol), and amino acid residues involved in interactions of B3 against each target are also shown in Table 1.

### 3.2. Effect on Thermal Latency

Thermal hyperalgesia was determined utilizing the hot plate method. B3 at 10 mg/kg body weight significantly increased (*p* < 0.05) and (*p* < 0.01) thermal latency on day 10th and 14th, respectively. B3 at 20 mg/kg body weight also significantly increased (*p* < 0.001) thermal latency on day 10th and 14th, as shown in Figure 7.

### 3.3. Effect on Mechanical Allodynia

Mechanical allodynia was determined with the Von Frey method. B3 (10 and 20 mg/kg body weight), significantly (*p* < 0.001) increased pain perception on day 10th and 14th, respectively, as shown in Figure 8.

### 3.4. Effect on Paw Deformation

The effect of B3 on paw deformation was also investigated. The footprints of the paws in the saline group were normal. In case of CCI, the paw posture was severely affected and the animal was unable to place it normally on a smooth surface. B3 treatment resulted in attenuating the CCI-induced paw deformation, as shown in Figure 9.

### 3.5. Effect on LPO and Nitric Oxide

LPO and iNOS were markedly elevated (*p <* 0.001) in CCI group as shown in Table 2 (a and b). Treatment with B3 significantly decreased the level of LPO and iNOS by (*p <* 0.001) in both the sciatic nerve and the spinal cord.

### 3.6. Effect on Oxidative Stress Enzyme

B3 was also investigated for its effect on certain oxidative stress related enzymes like GSH and GST, as shown in Table 2 (a and b). The low levels of GSH and GST were noticed (*p <* 0.001) in cases of CCI. B3 administration significantly reduced the oxidative stress (*p <* 0.001) by increasing the expression of both GSH and GST in the sciatic nerve and spinal cord, which could play a role in ameliorating CCI-induced painful effects.

### 3.7. Effect of Histopathological Changes (H and E and Immuno-Histochemical Changes)

H&E and IHC staining of the sciatic nerve showed a well-organized cellular structure, with no infiltration and intact intracellular spaces without edema in the sham group, as shown in Figure 10. In CCI group the injury of sciatic nerve was associated with various types of nerve damage as well as increase in the cellular structure, infiltration, increase in intracellular spaces, disorganized edema pattern because of the damage, as indicated by the arrows in Figure 10. Treatment with B3 (20 mg/kg body weight) reversed the constriction induced pathological changes.

### 3.8. Effects on Inflammatory Marker (IL-6)

The effects of B3 on the expression of IL-6 in the sciatic nerve and spinal cord, as shown in Figure 11, was also investigated. IL-6 was found elevated in CCI group (*p <* 0.001 vs. sham). B3 treatment decreased the elevated level of IL-6 (*p* < 0.01 vs. CCI) after 14 days of treatment in both the sciatic nerve and spinal cord.

## 4. Discussion

The chronic constriction injury-induced neuropathic pain (NP) model is a reproducible model for determining the effects of any pharmacological agents [20,21,22]. It has been already established and well explained [2] that the level of inflammatory mediators including IL-6, TNF, and IL-10 markedly increase in a damaged or constricted sciatic nerve. The raised expression level of COX-2 is another marker involved in the mediation and development of hyperalgesia and mechanical allodynia [23,24]. In a recently published study, it has been established that iNOS is also involved in the development and maintenance of CCI-induced NP [25]. The current research is based on the protective effect of a synthetic compound (B3) on neuro-inflammation induced by CCI. The results reveal that the administration of B3 in 10 and 20 mg/kg for 14 days significantly reverse the behavioral changes, paw deformation, and disturbance in oxidative stress level and suppress the over-expression of inflammatory mediators with special focus on IL-6 in CCI-induced NP. In order to determine the effect of B3 on NP, various approaches including computational, behavioral, and molecular were utilized.

In the computational analysis, the ligand was docked against various targets as mentioned earlier and the molecular interactions were obtained by visualizing the docked ligand and target through DSV 2016. Molecular interactions play an important role in stabilizing the ligand target complex. These molecular interactions are in the form of hydrogen bonds and hydrophobic interactions provide stability to the target ligand complex [26]. The interactions (hydrogen bonds) of ligand (B3) with IL-6 (3 H bonds), COX-2 (2 H bonds,) and iNOS (1 H bond) as shown in (Table 1) reflect the affinity of the ligand with the mentioned targets and hence provide a base that the ligand may be preceded for in vivo experimentation in the animal model. 

Sciatic nerve ligation-induced NP was characterized by sensory disturbances like mechanical and thermal hyperalgesia [27]. In our research, we found that the response of the rats to mechanical and thermal stimuli was significantly elevated as shown in Figure 8 and Figure 9, which is in line with the findings of previous studies [28]. The neuro-protective effect of B3 increases the threshold of pain and reverses the hyper responsiveness in rats to attenuate hyperalgesia.

In most diseases, oxidative stress is the underlying pathophysiology associated with worsening the progression of the disease [29]. As shown in Table 2, nerve ligation results in the elevation of iNOS and LPO, which are responsible for the development of oxidative stress by generating free radicals and damaging the nerve tissue, resulting in the development of hyperalgesia and other painful consequences. The reduction level of GST and GSH in Table 2 also complies with the previously published data of the CCI-induced NP model [30]. Hence, we can say that the antioxidant nature of B3 in treated groups played a critical role in preventing oxidative stress and reversing the abnormal disease pattern.

It has been known that the spinal cord is in direct connection with the sciatic nerve, so damage to sciatic nerve may induce inflammation by increasing the level of inflammatory mediators and initiating the system of reactive oxygen species [31]. In a previous study, it was found that microglia are not required to induce mechanical pain hypersensitivity in female mice. Rather, female mice use their adaptive immune cells to achieve the same level of hypersensitivity. This recommended the use of female mice in the pain research [32]. The inflammation of the spinal cord results in the production of arcolein, a reactive aldehyde which is particularly involved in promoting free radicals and depleting protective endogenous antioxidants of the body [32,33]. Several agents have been researched so far for attenuating the process of reactive oxygen species and thus preventing the development of symptoms in the spinal cord following CCI. The stable binding mode against IL-6, the correction of sensory behavior (mechanical and thermal hyperalgesia), paw deformation (Figure 7), and the prevention of oxidative stress were the reasons for utilizing the molecular approach in determining the possible mechanistic pathway for the neuro-protective nature of B3. Hence, the tissue samples of sciatic nerve and spinal cord were processed for ELIZA to determine the expression of IL-6 in the sham treatment and CCI groups. The up regulation of IL-6 in CCI has been extensively studied in pro-inflammatory cytokines. Minocycline, beside its antibacterial nature, has recently gained attention of researchers for its anti-nociceptive and anti-inflammatory potential by down regulating the level of IL-6 and correcting abnormal disease progression. Pregabalin is also studied for its potential in lowering the over expressed state of IL-6 and stopping disease progression [34,35]. In our study, the effects of B3 are consistent with the already developed model of CCI-induced neuropathic pain as shown in Figure 11. Hence, it is concluded that B3 treatment attenuates behavioral, biochemical, and inflammatory changes induced by CCI in the rat model.

## 5. Conclusions

The results demonstrate that B3 possess anti-nociceptive and anti-hyperalgesic effects and thus minimizes the pain perception in CCI-induced neuropathic pain. It corrects the behavioral changes, paw deformation/posture, elevates the level of protective GST and GSH and down regulates oxidative stress markers (iNOS and LPO). It reverses the CCI-induced pathological changes, confirmed by H&E staining and IHC staining of the sciatic nerve. Hence, we can postulate that the possible underlying mechanism for the neuro-protective effect of B3 may be mediated through IL-6, which is found to be up regulated in CCI and was normalized in 14 days post treatment by B3. Further investigations are needed to determine its pharmacokinetics profile and stability to add an active moiety to the management of neuropathic pain for clinical applications.

## Figures and Tables

**Figure 1 brainsci-10-00731-f001:**
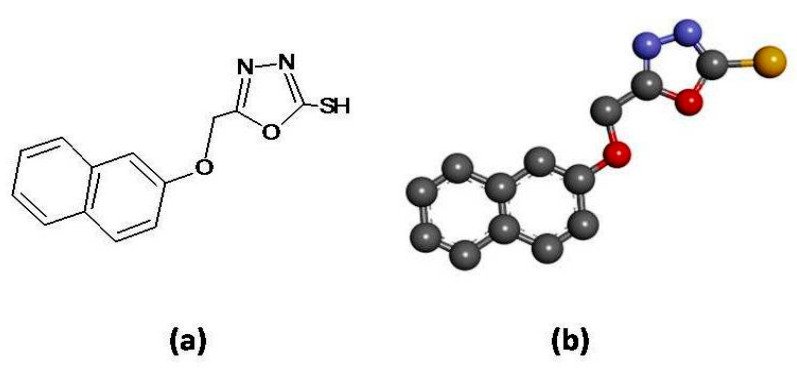
(**a**) Represents chemical structure of 5-[(naphthalen-2-yloxy)methyl]-1,3,4-oxadiazole-2 thiol (B3), (**b**) Represents 3D structure of 5-[(naphthalen-2-yloxy)methyl]-1,3,4 oxadiazole-2-thiol drawn through Discovery studio Visualizer client 2016 (B3).

**Figure 2 brainsci-10-00731-f002:**
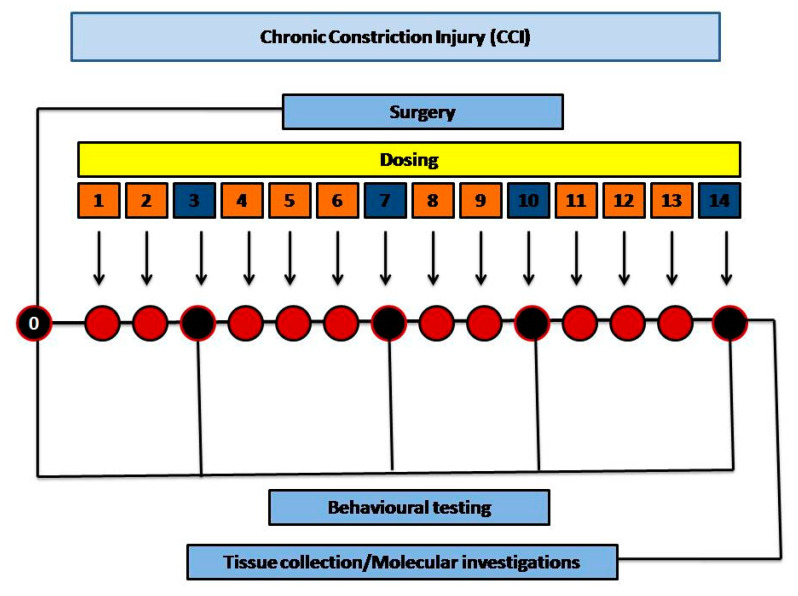
Schemeof the experimental work.

**Figure 3 brainsci-10-00731-f003:**
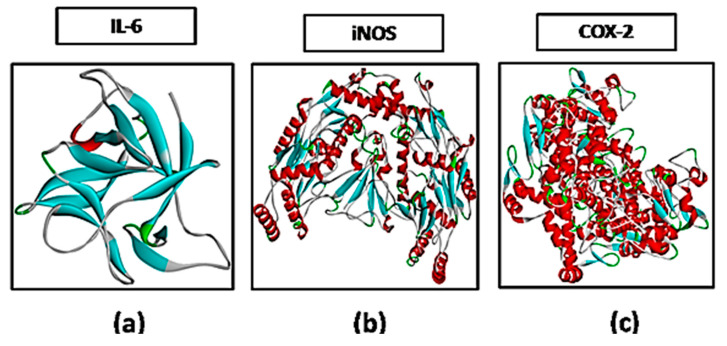
Representation of 3D structures of (**a**) interleukin-6 (IL-6), (**b**) inducible nitric oxide synthase (iNOS) and (**c**) cyclooxygenase-2 downloaded from protein data bank.

**Figure 4 brainsci-10-00731-f004:**
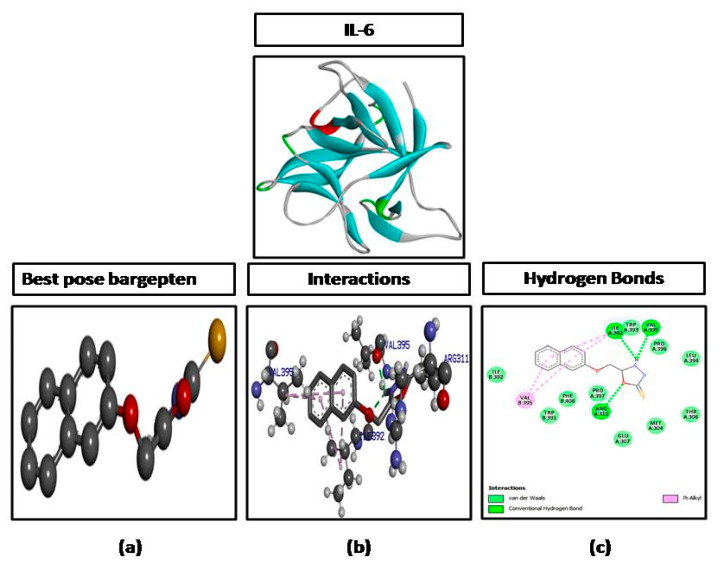
Represents docking of 5-[(naphthalen-2-yloxy)methyl]-1,3,4-oxadiazole-2-thiol (B3) against interleukin-6 (IL-6). (**a**) Represent best pose (**b**) represent interactions and (**c**) represent hydrogen bonds.

**Figure 5 brainsci-10-00731-f005:**
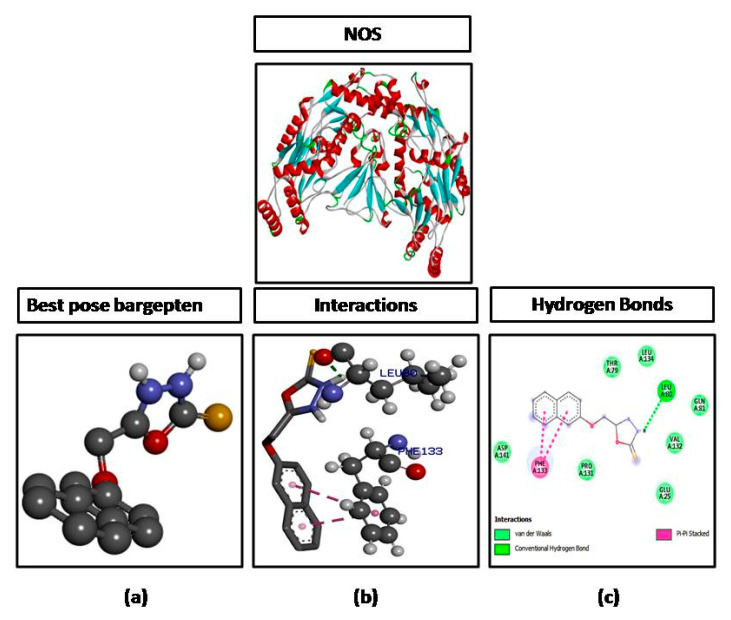
Represents docking of 5-[(naphthalen-2-yloxy)methyl]-1,3,4-oxadiazole-2-thiol (B3) against nitric oxide synthase (NOS). (**a**) Represent best pose, (**b**) representinteractions and (**c**) represent hydrogen bonds.

**Figure 6 brainsci-10-00731-f006:**
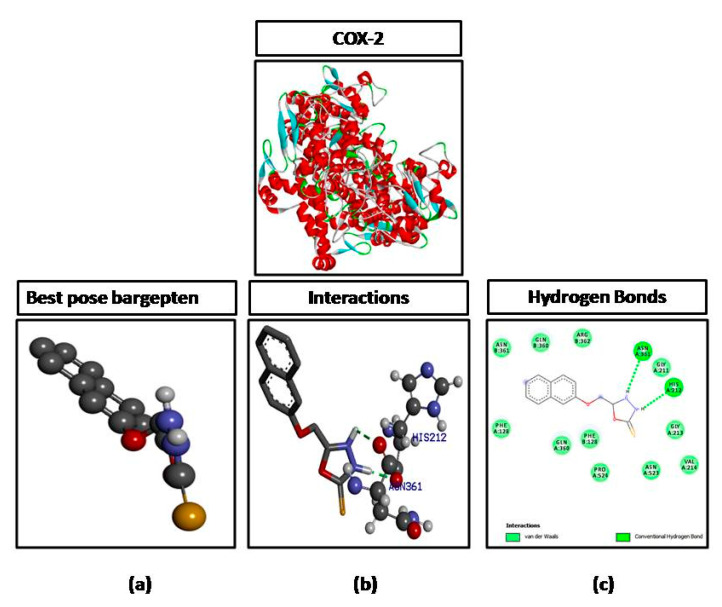
Represents docking of 5-[(naphthalen-2-yloxy)methyl]-1,3,4-oxadiazole-2-thiol (B3) against cyclooxygenase-2 (COX-2). (**a**) Represents best pose (**b**) represent interactions and (**c**) represent hydrogen bonds.

**Figure 7 brainsci-10-00731-f007:**
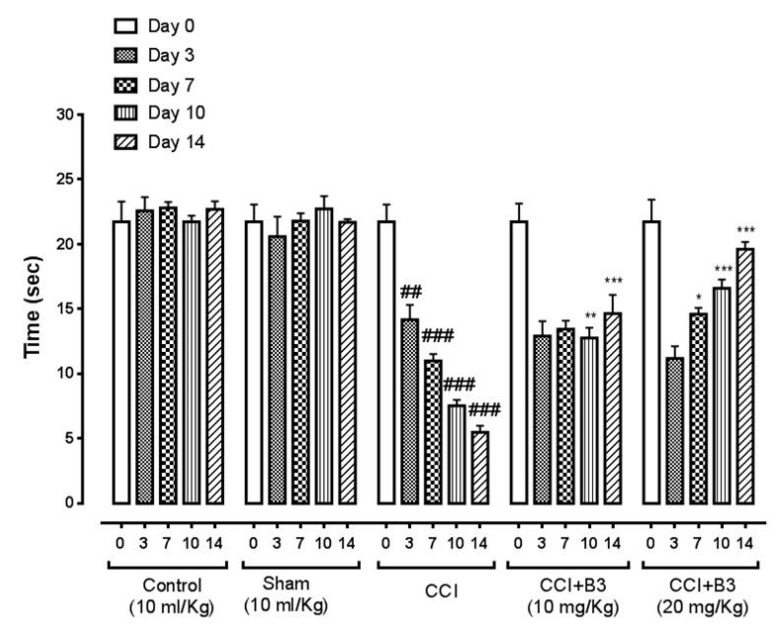
Bar graph represents the effect of 5-[(naphthalen-2-yloxy)methyl]-1,3,4 oxadiazole-2 thiol (B3) on thermal hyperalgesia on 0, 3, 7, 10 and 14th day (Data is expressed as mean ± SEM, *n* = 6. * *p* < 0.05, ** *p* < 0.01, *** *p* < 0.001 vs. sham, ### *p* < 0.001 vs. CCI Two-way ANOVA with post-hoc Bonferroni multiple comparisons test).

**Figure 8 brainsci-10-00731-f008:**
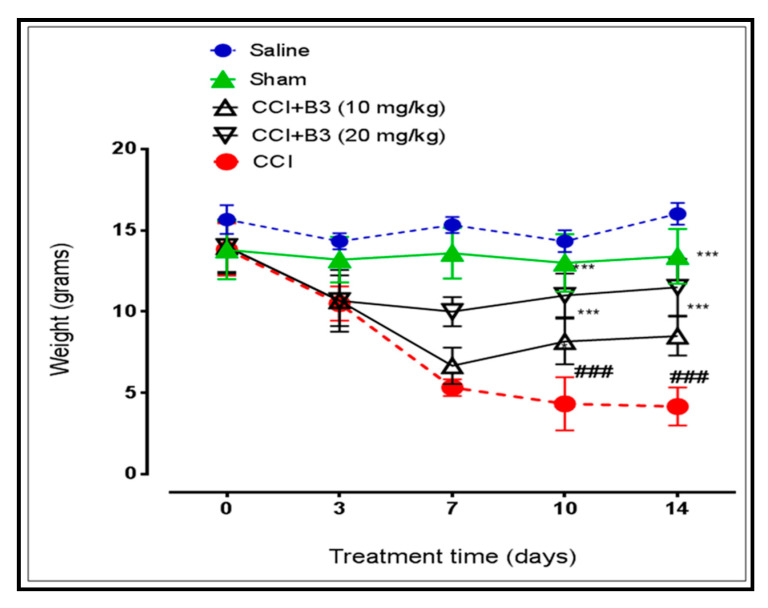
Line graph represents the effect of 5-[(naphthalen-2-yloxy)methyl]-1,3,4 oxadiazole-2 thiol (B3) on mechanical allodynia on 0, 3, 7, 10 and 14th day (Data is expressed as mean ± SEM, *n* = 6. *** *p* < 0.001 vs. sham, ### *p* < 0.001 vs. CCI. Two-way ANOVA with post-hoc Bonferroni multiple comparisons test).

**Figure 9 brainsci-10-00731-f009:**
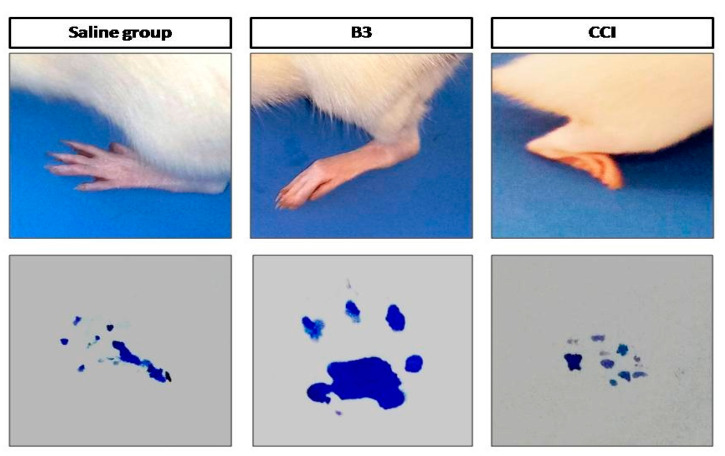
Represents the effect of 5-[(naphthalen-2-yloxy)methyl]-1,3,4 oxadiazole-2 thiol (B3) on paw deformation in chronic constriction injury.

**Figure 10 brainsci-10-00731-f010:**
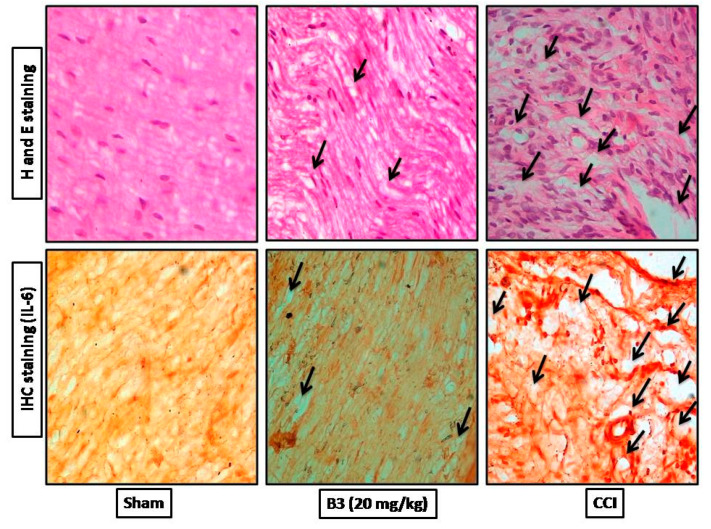
Histological examinations of rat sciatic nerve on the 14th day after surgery through H&E staining and immuno-histopathological (IHC) staining. The sham group showed organized cellular pattern without edema and infiltration. CCI group shows worse cellular changes, increase in cellular spaces and inflammatory cell infiltration. B3 showed minimal infiltration and cellular edema. The arrows show the areas of degradation caused by CCI. Magnifications, 40×.

**Figure 11 brainsci-10-00731-f011:**
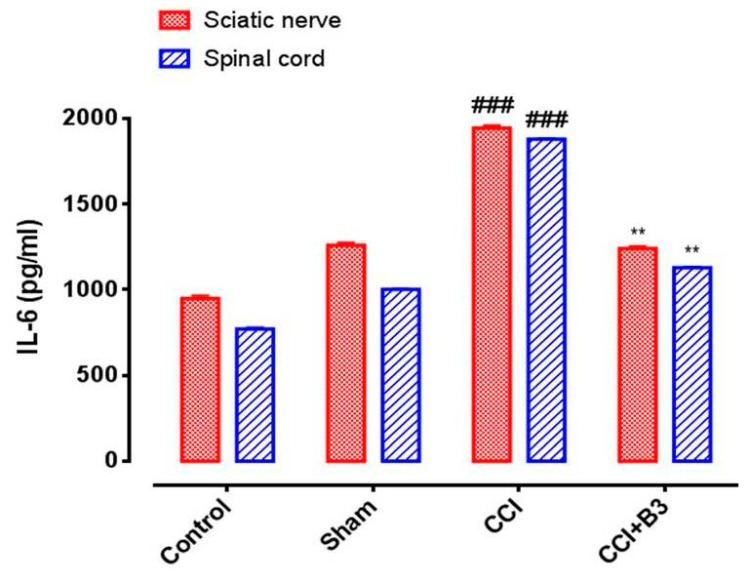
Bar graph represents the effect of 5-[(naphthalen-2-yloxy) methyl]-1,3,4-oxadiazole-2 thiol (B3) on the expression level of interleukin-6 (IL-6) in sciatic nerve and spinal cord (Data expressed as mean ± SEM, *n* = 6. ** *p* < 0.01 vs. sham group, ### *p* < 0.001 vs. CCI. One-way ANOVA post-hoc Bonferroni multiple comparisons test).

**Table 1 brainsci-10-00731-t001:** Represents targets including nitric oxide synthase (NOS), interleukin-6 (IL-6), and cyclooxygenase-2 (COX-2), score (Kcal/mol), no of hydrogen bonds, amino acid residues interpreted through Biovia Discovery Studio Visualizer 2016.

Compounds	Interleukin-6			iNOS			COX-2		
	Score	H-Bond	Bonding Residues	Score	H-Bond	Bonding Residues	Score	H-Bond	Bonding Residues
B3	−9.2	03	ARG 311 ILE 392 VAL395	−6.6	01	LEU 80	−7.9	02	ASN 361 HIS 212

Histidine = HIS, Arginine = ARG, Isoleucine = ILE, Leucine = LEU, Phenylalanine = PHE, Serine = SER, Tryptophan = TRP, Threonine = THR, Tyrosine = TYR, Valine = VAL.

**Table 2 brainsci-10-00731-t002:** (a) Shows the effect of B3 on oxidative enzymes of sciatic nerve, (b) Shows effect of B3 on oxidative enzymes of spinal cord.

(a)	**Group**	**GSH**	**GST**	**iNOS**	**LPO**
	Saline	83.22 ± 4.2	73.88 ± 2.3	43.22 ± 1.2	91.13 ± 3.1
	Sham	77.26 ± 1.2	69.78 ± 2.4	46.41 ± 2.6	87.16 ± 3.1
	CCI	19.42 ± 2.2 ###	17.33 ± 1.4 ###	105.12 ± 1.7 ###	322.26 ± 1.8 ###
	B3	55.23 ± 2.1 ***	66.33 ± 1.0 ***	75.20 ± 2.1 **	270.76 ± 2.4 **
(b)	**Group**	**GSH**	**GST**	**iNOS**	**LPO**
	Saline	66.12 ± 3.2	48.11 ± 1.1	44.15 ± 2.4	88.23 ± 3.3
	Sham	63.22 ± 1.2	49.26± 2.8	51.31 ± 1.0	78.26 ± 1.8
	CCI	8.41 ± 1.5 ###	10.53 ± 3.4 ###	153.82 ± 1.3 ###	210.68 ± 1.6 ###
	B3	38.87 ± 3.5 ***	25.73 ± 1.0 **	51.32 ± 3.0 ***	121.76± 2.8 ***

Data expressed as mean ± SEM, *n* = 6. ** *p <* 0.01 and *** *p <* 0.001 vs. sham group, ### *p <* 0.001 vs. CCI. One-way ANOVA post-hoc Bonferroni multiple comparisons test.

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
