# Peer review of "1,3,4-Oxadiazole Derivative Attenuates Chronic Constriction Injury Induced Neuropathic Pain: A Computational, Behavioral, and Molecular Approach"

_brainsci, 2020, doi:10.3390/brainsci10100731_

Round 1

Reviewer 1 Report

The Authors highlight the possible use of 1,3,4-Oxadiazole derivative in a widely studied model of neuropathic pain. 

The paper is well conducted and the data are sufficiently presented. 

The quality of the histological figures should be improved. 

There are several typos that need to be fixed. 

Some data in the figure is not properly presented. I.E. behavior as Fig. 11or fig 8 the CCI group should be before the treated group (i.e. sham, CCI and relative treatments.)

Statistical analysis should be revised F values should be provided. 

Some important paper in the literature highlighting the role of neuroinfammatory processes in the spinal cord and sciatic nerve of neuropathic mice should be provided. (i.e. Hains and waxman 2006 J. Neurosci, Guida et al., 2012, Molecular pain). 

Gender differences should be provided or at least discussed (Sorge et al., 2015 Nat Neurosci.; Boccella et al., 2019 FASEB-J). 

English should be revised 

Author Response

Reponses to comments of Reviewer 1

It is to confirm that Dr. Muhammad Faheem and Dr. Muhammad Zahoor are the corresponding authors of the article. As per the Respected reviewer following changes were made;

Suggestion 1: The quality of the histological figures should be improved. 

Response: Dear Editor, The quality of the histopathological images has been improved.

Suggestion 2: There are several typos that need to be fixed. 

Response: Dear Editor, The document is carefully evaluated and all the typing mistakes were fixed.

Suggestion 3: Some data in the figure is not properly presented. I.E. behavior as Fig. 11or fig 8 the CCI group should be before the treated group (i.e. sham, CCI and relative treatments.)

Response: Dear Editor, The data presented in Figure 8 and Figure 11 has been arranged as per reviewer comments i.e. control, sham CCI, and Treatment groups.

Suggestion 4: Statistical analysis should be revised F values should be provided. . 

Response: Dear Editor, as we are not expert in this field being biochemist/pharmacist it is difficult for us perform further complicated analysis. Hope worthy reviewer will understand the situation

Suggestion 5: Some important paper in the literature highlighting the role of neuroinfammatory processes in the spinal cord and sciatic nerve of neuropathic mice should be provided. (i.e. Hains and waxman 2006 J. Neurosci, [1]). 

Response: Dear Editor, the discussion section has been updated using the suggested paper (Hains and waxman 2006 J. Neurosci, Guida et al., 2012, Molecular pain)

Suggestion 6: Gender differences should be provided or at least discussed (Sorge et al., 2015 Nat Neurosci.;Boccella et al., 2019 FASEB-J). 

Response: Dear Editor, the impact of experimental animal gender on the pain research has added to the discussion section.

Suggestion 7: English should be revised

Response: The manuscript has been thoroughly revived for the correction of mistakes, spacing and the English language is been improved in the manuscript.

Reviewer 2 Report

The article "1,3,4-Oxadiazole derivative attenuates chronic constriction injury induced neuropathic pain. A computational, behavioral and molecular approach" by Faheem et al. study the effect of put forward the importance of 1,3,4-Oxadiazole derivative (B3) in neuropathic pain using several techniques like molecular docking, behavior and histopathology. The manuscript lacks the use of the standard drug, appropriate use of statistics and proper representation of data. It needs a thorough grammar check and spelling check. And can be considered after the Major revision.

Major comments:
1) The obtained average latency should be converted to percent of maximal possible effect (% MPE) and then reported for a better understanding of the percent inhibition of pain.

2) The statistics used one way ANOVA for the analysis of the behavioral study is inappropriate. Authors should reanalyze the data using either two way ANOVA with Bonferroni's multiple comparison test. Since the same rats were treated for 14 days and were assessed at 0, 3, 7, 10 and 14 days, the use of two way ANOVA is recommended.

3) Since authors didn't use any standard drug to compare the effects of B3 with, it is recommended to use some standard drug as a reference in the study and compare the effect of B3 with the reference standard with %MPE.

4) The authors mention the use of the primary antibodies (anti-TNF-α, anti-COX-2, and anti-NF-κb) in the Materials and methods section Line: 180. However, in Figure 10, the authors do not specify any antibody. It is not at all acceptable that the outcome of three antibodies in one focus. Moreover, it is confusing if three antibodies were used then there should be three IHC observation panels depending upon the source of antibodies which is not the case in the present manuscript. Please add the effect of SHAM, CCI and B3 treatment on all three IHC components individually (anti-TNF-α, anti-COX-2, and anti-NF-κb). Also authors need to add the specificity of these antibodies (eg. raised in which species and acting against what species) along with detailed source.

Minor Comments:

1) Materials and Methods: Line 70 and 83: There is a contradiction between the two statements. What was the dose used for B3 10, 20, 30 25 or 35 mg/kg. Its mentioned differently in both the statements. Please justify the difference in the doses mentioned.

2) Materials Line 69: Dihydrodithiobisnnitro benzoic acid should be Dihydrodithiobisnitro benzoic acid. Xylaxine should be Xylazine.

3) It shall be a good addition to the manuscript if authors can report the footprint length in the revision. This can make readers understand the difference between control, CCI and B3 treated.

4) Please justify the rationale behind using the spinal cord for oxidative enzymes in the discussion.

Author Response

Reponses to comments of Reviewer 2

General comments:

The manuscript lacks the use of the standard drug, appropriate use of statistics and proper representation of data.

Response: Dear Editor, The standard drug was not utilized in the study because the comparison of the test group is directly made with CCI group rather than comparing it with standard, secondly as the model represent an intentionally induced crush injury for which the research lacks a specified standard drug. Some paper explains the role of pregabalin but again is not considered to be the standard drug in crush injury nor is it indicated in clinical practice for neurological crush injuries.

Comment 1: The obtained average latency should be converted to percent of maximal possible effect (% MPE) and then reported for a better understanding of the percent inhibition of pain.

Response: Dear Editor, The average thermal latency has been converted to maximal possible effect (% MPE) for day 0, 3rd, 7th 10th and 14th and is given below. The percentage pain inhibition in treated group by B3 is 0.0%, 21.15%, 24.65%, 54.81% and 71.19% on day 0, 3rd, 7th 10th and 14th day respectively which is reflecting the pain killing effect of B3 in CCI-induced neuropathic pain

Comment 2: The statistics used one way ANOVA for the analysis of the behavioral study is inappropriate. Authors should reanalyze the data using either two way ANOVA with Bonferroni's multiple comparison test. Since the same rats were treated for 14 days and were assessed at 0, 3, 7, 10 and 14 days, the use of two way ANOVA is recommended.

Response: Dear Editor, although we have revised the section however, we are not expert in this field being biochemist/pharmacist it is difficult for us perform further complicated analysis. Hope worthy reviewer will understand the situation.

Comment 3: Since authors didn't use any standard drug to compare the effects of B3 with, it is recommended to use some standard drug as a reference in the study and compare the effect of B3 with the reference standard with %MPE.

Response: Dear Editor, %MPE has already provided as mentioned above. Dear Editor, The standard drug was not utilized in the study because the comparison of the test group is directly made with CCI group rather than comparing it with standard, secondly as the model represent an intentionally induced crush injury for which the research lacks a specified standard drug. Some paper explains the role of pregabalin but again is not considered to be the standard drug in crush injury nor is it indicated in clinical practice for neurological crush injuries.

Maximal percentage response. 

Group

Number of Days

(Percentage pain inhibition)

0 day

3rd day

7th day

10th day

14th day

Control vs. CCI

0

37.2

51.75

62.55

75.77

Sham vs. CCI

0

31.0

49.54

66.96

71.36

Sham vs. B3

0

45.0

33.02

27.00

9.67

CCI vs. B3

0

21.15

24.65

54.81

71.19

Comment 4: ) The authors mention the use of the primary antibodies (anti-TNF-α, anti-COX-2, and anti-NF-κb) in the Materials and methods section Line: 180. However, in Figure 10, the authors do not specify any antibody. It is not at all acceptable that the outcome of three antibodies in one focus. Moreover, it is confusing if three antibodies were used then there should be three IHC observation panels depending upon the source of antibodies which is not the case in the present manuscript. Please add the effect of SHAM, CCI and B3 treatment on all three IHC components individually (anti-TNF-α, anti-COX-2, and anti-NF-κb). Also authors need to add the specificity of these antibodies (eg. raised in which species and acting against what species) along with detailed source.

Response: Dear Editor, The only primary anti-IL-6 antibody has been utilized for immunohistopathological analysis and the images reveal the IHC of single anti body. This was a typing error and has been corrected.

Minor Comments:

Minor Comment 1: Materials and Methods: Line 70 and 83: There is a contradiction between the two statements. What was the dose used for B3 10, 20, 30 25 or 35 mg/kg. Its mentioned differently in both the statements. Please justify the difference in the doses mentioned.

Response: Dear Editor, The dose of the test drug is 10 and 20 mg/kg body weight and is corrected at every point in the manuscript. it is to clear that only animal group of 20 mg/kg body weight has been processed for histological studies

Minor Comment 2: Materials Line 69: Dihydrodithiobisnnitro benzoic acid should be Dihydrodithiobisnitro benzoic acid. Xylaxine should be Xylazine.

Response: Dear Editor, The spelling of both Dihydrodithiobisnitro benzoic acid and Xylazine has been corrected in material section of the manuscript.

Minor Comment 3: It shall be a good addition to the manuscript if authors can report the footprint length in the revision. This can make readers understand the difference between control, CCI and B3 treated.

Response: Dear Editor, The foot print length is calculated and is given in following table

Group

Length (cm)

Width (cm)

Sham 

3.8

2.9

B3

3.0

2.4

CCI

1.8

1.7

Minor Comment 4: Please justify the rationale behind using the spinal cord for oxidative enzymes in the discussion

Response: Dear Editor, The ROS and oxidative stress mechanism has been widely known in case of neurological injury to sciatic nerve, spinal cord or brain. The rationale of the ROS in spinal cord has been justified and highlighted in discussion with addition of references of suitable work

Round 2

Reviewer 1 Report

The Authors addressed the comments

and the paper is suitable for publication

Reviewer 2 Report

The revision submitted by Faheem and collegues for manuscript entitled ''1,3,4-Oxadiazole derivative attenuates chronic constriction injury induced neuropathic pain. A computational, behavioral and molecular approach'' is very well revised and addresses all the comments raised by Reviewer. The manuscript can be accepted for publication in Brain Sciences.